# OpenReview forum: "Next-Gen CAPTCHAs: Leveraging the Cognitive Gap for Scalable and Diverse GUI-Agent Defense"
_ICML.cc/2026/Conference — ICML 2026 regular_

### Official Review · Reviewer_XThw · 2026-02-13

**Soundness:** 3
**Presentation:** 3
**Significance:** 3
**Originality:** 3
**Overall Recommendation:** 4
**Confidence:** 4

**Summary:**

The paper introduces next-generation CAPTCHAs designed to address the growing threat posed by GUI-enabled agents, which have rendered traditional CAPTCHA mechanisms increasingly ineffective. CAPTCHAs are widely used to distinguish humans from AI systems and to mitigate automated abuse by bots. However, modern GUI-enabled agents, powered by MLLMs and capable of interactive task execution, present a significant challenge to these conventional defenses.

To counter this threat, the authors propose a GUI-agent defense framework that generates interactive challenges specifically designed to resist GUI-based agents while remaining easily solvable for humans. The paper presents several new CAPTCHA families, developed based on identified cognitive and decision-making gaps between humans and AI agents. Experimental results demonstrate that these CAPTCHAs are user-friendly for humans but cause GUI-enabled agents to fail with high probability. Additionally, ablation studies uncover several insightful findings that may inform and guide the future design of robust CAPTCHA systems.

**Compliance With Llm Reviewing Policy:**

Affirmed.

**Final Justification:**

Fully resolved - My concerns have been adequately addressed, and I have a positive view of the paper.

**Key Questions For Authors:**

N/A

**Limitations:**

As mentioned earlier, it would be good to include a discussion of limitations, e.g., regarding adaptive adversaries and the potential evolution of agent capabilities.

**Strengths And Weaknesses:**

This paper addresses a timely and important problem: the growing ineffectiveness of traditional CAPTCHAs in the face of GUI-enabled agents powered by multimodal large language models. The paper responds to this challenge by proposing a new framework for generating interactive CAPTCHAs that are specifically designed to exploit cognitive and decision-making gaps between humans and current AI agents. Overall, the work is well motivated and relevant to both the machine learning and security communities.

From a soundness perspective, the paper is largely convincing in its empirical claims. The proposed CAPTCHA families are grounded in documented gaps between human and agent cognition, which provides a principled design foundation rather than an ad hoc construction. The experimental evaluation demonstrates that the CAPTCHAs are readily solvable by humans while causing high failure rates for GUI-enabled agents, and the inclusion of ablation studies strengthens the empirical case by identifying which components most contribute to robustness. The authors appear careful in their analysis and do not overstate the experimental findings. **However**, the formal modeling of GUI-enabled agents as an extended POMDP contributes less than it could. The formulation is semi-formal and does not yield theoretical guarantees or meaningful analytical leverage. It neither supports formal proofs nor establishes provable security margins, limiting its utility. Additionally, the evaluation does not consider adaptive attacks. The absence of experiments or analysis involving adaptively fine-tuned or designed agents raises concerns about long-term robustness. Some of the identified cognitive decision gaps, such as G2 and G5, seem contingent on the current limitations of AI systems rather than on fundamental barriers. There is no evidence that these gaps are intrinsically difficult for AI to overcome, which makes it unclear how resilient the proposed approach would be as models continue to improve.

In terms of presentation, the paper is generally clear and well-structured.

The significance of the work is substantial. Automated abuse prevention is a critical and widespread problem, and the rise of powerful GUI-enabled agents creates an urgent need for new defensive mechanisms. The paper could influence both future research and practical deployments. Even if the specific CAPTCHA families eventually require adaptation, the broader framework may provide a foundation for iterative defenses against increasingly capable AI systems. While the scope is somewhat specialized, the practical implications are broad given the ubiquity of CAPTCHA-based defenses in online systems.

Regarding originality, the paper demonstrates meaningful novelty primarily in its design and application rather than in formal theory. The systematic exploitation of human–agent decision gaps represents a thoughtful and timely contribution.

In summary, this paper makes a strong empirical contribution to an important and emerging problem. Although weaknesses are present, the work is a valuable step toward future CAPTCHA design.

---

> ### Author Rebuttal · Authors · 2026-03-31
>
> We thank the reviewer for the insightful comments and constructive suggestions. We have carefully considered all the points raised and will revise the manuscript accordingly. Below, we provide a point-by-point response to each comment.
>
> >**W1: Utility of the extended POMDP formulation.**
>
> We appreciate the comment. We clarify that our extended POMDP formulation is not intended to provide formal proofs or provable security margins. Its purpose is more modest: to make explicit the GUI-agent CAPTCHA loop: *observation, internal state, deliberation, browser action, verifier reward/cost* and to organize the empirically observed failure modes that motivate the G1–G5 taxonomy.
>
> We will revise Sec.4.1 to make this role explicit, tone down any implication of formal security guarantees, and present the formulation as a conceptual framework that supports the design taxonomy rather than as a model intended to support formal guarantees.
>
>
> >**W2: Adaptive-attack concern.**
>
>
> Thanks for the suggestion. To further study adaptive attack risk, we additionally evaluate a task-type-hint baseline on the full 519-puzzle test set using Gemini-3-Flash-High under the same Browser-Use protocol as in the paper. For each CAPTCHA family, we provide the attacker agent with a family-level strategy hint. Specifically, we ask Gemini-3-Pro-High to synthesize one general UI-based solving strategy for each family, describing what the agent can observe and do on the webpage, given the corresponding generator logic, frontend interaction code, server-side verification logic, and browser-agent task description. We view this as a lightweight adaptive baseline based on family-aware prompting and test-time task guidance. This captures a practically relevant attack setting, since prompt-level adaptation is a widely used and often effective strategy for modern LLM agents.
>
>
> Under this setting, Pass@1 increases from 3.2% to 6.4% overall, but the gains are highly uneven across families, and many families remain at 0%. In particular, most of the overall improvement comes from Red_Dot, while the broader benchmark remains difficult for the agent, suggesting that most CAPTCHA families remain difficult under prompt adaptation.
>
> |Puzzle Type|Baseline Accuracy (%)|Adaptive Accuracy (%)|Δ Accuracy (pp)|
> |---|:---:|:---:|:---:|
> |Overall|3.2|6.4|+3.2|
> |3D_Viewpoint|0.0|0.0|0.0|
> |Backmost_Layer|5.0|0.0|-5.0|
> |Box_Folding|10.0|5.0|-5.0|
> |Color_Counting|0.0|5.0|+5.0|
> |Dice_Roll_Path|15.0|20.0|+5.0|
> |Dynamic_Jigsaw|0.0|0.0|0.0|
> |Hole_Counting|0.0|5.0|+5.0|
> |Illusory_Ribbons|0.0|0.0|0.0|
> |Layered_Stack|0.0|0.0|0.0|
> |Mirror|9.0|0.0|-9.0|
> |Multi_Script|5.0|0.0|-5.0|
> |Occluded_Pattern_Counting|0.0|5.0|+5.0|
> |Red_Dot|15.0|100.0|+85.0|
> |Rotation_Match|0.0|0.0|0.0|
> |Shadow_Direction|0.0|0.0|0.0|
> |Shadow_Plausible|0.0|12.5|+12.5|
> |Spooky_Circle|20.0|5.0|-15.0|
> |Spooky_Circle_Grid|0.0|0.0|0.0|
> |Spooky_Jigsaw|0.0|0.0|0.0|
> |Spooky_Shape_Grid|5.0|0.0|-5.0|
> |Spooky_Size|5.0|15.0|+10.0|
> |Spooky_Text|0.0|0.0|0.0|
> |Static_Jigsaw|0.0|0.0|0.0|
> |Structure_From_Motion|0.0|0.0|0.0|
> |Subway_Paths|0.0|0.0|0.0|
> |Temporal_Object_Continuity|0.0|0.0|0.0|
> |Trajectory_Recovery|0.0|0.0|0.0|
>
> We will include this experiment and explicitly clarify that G2/G5 are not claimed to be permanently hard for AI. Rather, they are currently effective defensive mechanisms, and procedural generation enables an iterative, continuously refreshed strategy as GUI agents evolve.
>
> Our claim is empirical rather than absolute: under realistic closed-API GUI-agent settings, Next-Gen CAPTCHAs preserve a substantial human–agent gap against strong frontier agents. The current evidence suggests that failures are not resolved simply by allocating more reasoning budget: increasing reasoning effort yields only modest gains, and changing the agent framework improves performance somewhat but still leaves overall success in the single digits. In many failures, the agent appears to localize the relevant region but still executes the wrong interaction, indicating that interactive perception-to-action bottlenecks remain significant under the current setup.
>
>
> >**L1: Discussion of adaptive adversaries and evolving agent capabilities.**
>
> Thanks for the suggestion. We will add a dedicated limitations paragraph covering adaptive adversaries, task-specific tuning, and the likely evolution of GUI-agent capabilities over time. We believe this makes the contribution more precise: the paper provides a strong empirical defense framework against current frontier GUI agents, not a claim of permanent or proof-based security.

---

> > ### Author Rebuttal · Reviewer_XThw · 2026-04-02
> >
> > Thanks for the response.

---

> > > ### Author Response · Authors · 2026-04-02
> > >
> > > Thanks and we are very glad to hear this. We will incorporate all these additional experiments, results, and clarifications into the revised paper. Thanks again for your constructive comments.

---

### Official Review · Reviewer_ZK81 · 2026-03-06

**Soundness:** 3
**Presentation:** 3
**Significance:** 3
**Originality:** 3
**Overall Recommendation:** 5
**Confidence:** 3

**Summary:**

This paper introduces Next-Gen CAPTCHAs, a systematic defense framework designed to secure the web against the advanced autonomous GUI-enabled agents. While recent advances of AI have enabled solving the traditional CAPTCHAs, the authors introduced a paradigm shift from static difficulty to dynamic cognitive divergence. By exploiting the cognitive gap, they built new CAPTCHAs that can well distinguish the AI and human.

**Compliance With Llm Reviewing Policy:**

Affirmed.

**Final Justification:**

The rebuttal sovled all of my concerns.

**Key Questions For Authors:**

See the weaknesses.

**Limitations:**

impact statement discussed.

**Strengths And Weaknesses:**

Strengths:

1.	A new family of CAPTCHAs is designed to defend the advanced AI system, which shows that it can detect most of the existing commercial systems.

2.	Interesting theoretical analysis of the cognitive gap strategy.

3.	The system uses rule-based verification, eliminating the costly and slow need for human manual labeling for new instances.

In general, I think this work is good. I prefer to accept this work. I only have some minor questions.

1.	Many of the "Next-Gen" tasks (e.g., Mirror, Shadow Direction, 3D Viewpoint) rely heavily on sophisticated visual and spatial reasoning. This could pose significant challenges for some users. Are they really human-easy?

2.	Because many tasks are dynamic and interactive (requiring motion cues or sequential reveals), they likely demand a more stable and higher-bandwidth internet connection than static image-grid CAPTCHAs.

---

> ### Author Rebuttal · Authors · 2026-03-31
>
> We are grateful to the reviewer for the constructive feedback. These comments will be incorporated into our revision and have been very helpful in strengthening the manuscript. We further provide detailed responses to each of the reviewer's questions below.
>
>
> >**W1: Human solvability.**
>
> Thanks for raising this point. In the current submission, the human pass study is a solvability check conducted with regular volunteers with CS/AI backgrounds. We tested 10 puzzles for each family, except 8 for Shadow Plausible, yielding 268 human trials in total. The overall average completion time across these trials is 31.1s per puzzle. Although a few families are more time-consuming, such as 3D Viewpoint (120.8s), Spooky Jigsaw (71.8s), Subway Paths (71.4s), and Multi Script (63.1s), the large majority of families are completed within one minute on average. Overall, these results suggest that human participants can solve most task families reliably after deliberation, with the main differences arising in time cost rather than solvability.
>
> To better assess usability beyond the original participant pool, we additionally conducted a broader user study with regular computer users without CS/AI backgrounds. In this study, the overall Pass@1 is 85.2%, which further suggests that the CAPTCHA families remain largely solvable for more general users.
>
> |Attribute|P1|P2|P3|P4|P5|P6|P7|P8|
> | --------- |:--|:--|:--|:--|:--|:--|:--|:--|
> |Education|Master|Bachelor|Primary School|Bachelor|Master|PhD|Bachelor|Primary School|
> |Age Group|20-30|20-30|10-20|80-90|50-60|50-60|20-30|10-20|
> |Expertise|Finance|Psychology|N/A|Business|Literature|Finance|Media|N/A|
>
> | |P1|P2|P3|P4|P5|P6|P7|P8|Overall|
> |------------|:------:|:------:|:------:|:------:|:------:|:------:|:------:|:------:|:--------:|
> |Pass@1 (%)|81.5|92.6|88.9|66.7|92.6|85.2|85.2|88.9|85.2|
>
> >**W2: Network requirements for dynamic tasks.**
>
>
> Thanks for the comment. We clarify that only a subset of CAPTCHA families rely on GIF-based or other non-static visual assets. To assess their practical network cost under current web conditions, we measured the average cold-load payload of these families, as summarized below:
>
> | GIF-based Task Type | Avg cold-load payload (MiB) |
> |---|:---:|
> | `Trajectory_Recovery` | 0.47 |
> | `Temporal_Object_Continuity` | 0.54 |
> | `Dynamic_Jigsaw` | 0.73 |
> | `Spooky_Circle_Grid` | 3.40 |
> | `Spooky_Shape_Grid` | 3.48 |
> | `Spooky_Text` | 3.67 |
> | `Spooky_Circle` | 3.86 |
> | `Spooky_Size` | 6.77 |
> | `Structure_From_Motion` | 7.73 |
> | `Spooky_Jigsaw` | 9.23 |
>
> These payloads are within a reasonable range for current web deployment, indicating that the GIF-based families are feasible under typical modern network conditions. For example, even the largest payload among the GIF-based families would require less than 1 second of transfer time at 100 Mbps in theory. Moreover, there remains room to further optimize the trade-off between user visibility and file size.
>
> We will add this discussion in the revision. In practice, a real deployment can (i) choose families conditioned on connection quality, (ii) pre-load lightweight assets when needed, and (iii) fall back to static / lower-bandwidth families for users with poor connectivity. Our intention is not that every deployment must use the most dynamic families, but that the framework offers a spectrum of agent-defensive tasks with different usability and systems trade-offs.

---

> > ### Author Rebuttal · Reviewer_ZK81 · 2026-04-02
> >
> > Thanks for the response.

---

> > > ### Author Response · Authors · 2026-04-02
> > >
> > > Thanks and we are very glad to hear this. We will incorporate all these additional experiments, results, and clarifications into the revised paper. Thanks again for your constructive comments.

---

### Official Review · Reviewer_Uzom · 2026-03-06

**Soundness:** 3
**Presentation:** 3
**Significance:** 3
**Originality:** 3
**Overall Recommendation:** 5
**Confidence:** 5

**Summary:**

This paper tackles a timely problem: existing CAPTCHAs are now trivially solved by frontier GUI agents (90%+ pass rates on Bingo-style puzzles), and the authors propose 27 new CAPTCHA families that exploit persistent human-agent "Cognitive Gaps" in perception, memory, and action execution. The key idea is to move away from decomposable logic puzzles toward interactive tasks where simply thinking harder doesn't help - agents fail not because they can't reason, but because they misfire on low-level action primitives like drag-and-drop. Results are striking: humans solve at 98.8% while best-in-class GPT-5.2 hits only 5.9%, and throwing more compute at the problem barely moves the needle.

**Compliance With Llm Reviewing Policy:**

Affirmed.

**Final Justification:**

The authors provided additional experiments on the human study as I asked. therefore, I am happy to increase the score

**Key Questions For Authors:**

Minor: It's good to notice the persistent agent-human cognitive gap and designed for that. I am actually a bit worried about scalable data synthesis for the CAPTCHAs. If they are scalable then the next-gen web agents can easily use that for training. How to fundamentally design CAPTCHAs tasks that hurt the general web performances. That's an open-question.

Minor: more philosiphy question: the web agent, tasked by a real human instead of auto crawling like a bot, should be considered as a bot and defended by website?




Typo:
Line 266.
so that -> So that?
or change equation (4) ending with ,

**Strengths And Weaknesses:**

# Pros:
1. Very timely and important question. The arms race framing is well motivated and the threat is real.
2. The 27 CAPTCHA families are thoughtfully designed and cover diverse interaction bottlenecks (G1-G5). The taxonomy is useful.
3. Experiments are comprehensive and the signals are clean - especially the cost-accuracy trade-off analysis showing economic asymmetry for attackers.

# Cons:
1. Human evaluation is a real concern. Some of these tasks (Illusory Ribbons, Spooky variants, Structure From Motion) look genuinely hard for humans too, not just agents. The paper claims 98.8% human pass rate but it's unclear how this was measured - sample size, participant pool, time limits. A proper human study is needed, not just a footnote.
I'd be happy to increase the score to Accept if authors provide detailed results here. The participants cant just be CS students or CS experts, they need to be regular computer users with or without higher education.

The below ones are more like discussions and wont affect how I rate this work.

Minor: It's good to notice the persistent agent-human cognitive gap and designed for that. I am actually a bit worried about scalable data synthesis for the CAPTCHAs. If they are scalable then the next-gen web agents can easily use that for training. How to fundamentally design CAPTCHAs tasks that hurt the general web performances. That's an open-question.

Minor: more philosiphy question: the web agent, tasked by a real human instead of auto crawling like a bot, should be considered as a bot and defended by website?

---

> ### Author Rebuttal · Authors · 2026-03-31
>
> We sincerely thank the reviewer for the constructive comments, which are helpful in improving the quality of our paper. We will carefully revise accordingly and include the suggested changes in the revised manuscript. In the following, we detail our responses to each of the questions raised.
>
> >**C1: Human evaluation concern.**
>
> Thanks for raising this. We clarify that the human pass results in the paper are based on a preliminary solvability study. Participants were regular computer users with CS backgrounds. We tested 10 puzzles per family for most families and 8 for Shadow Plausible, for a total of 268 human trials. We did not set an explicit time limit, and participants submitted their answers once they felt confident.
>
> To further assess human usability beyond the original participant pool, we conducted an additional small-scale user study with 8 participants from more diverse non-CS/AI backgrounds. Each participant attempted one instance from each of the 27 CAPTCHA families, yielding 216 human trials in total. The overall Pass@1 in this additional study is 85.2%, providing additional evidence that the benchmark remains largely solvable beyond the original CS/AI-background participant group.
>
> |Attribute|P1|P2|P3|P4|P5|P6|P7|P8|
> |---------|:--|:--|:--|:--|:--|:--|:--|:--|
> |Education|Master|Bachelor|Primary School|Bachelor|Master|PhD|Bachelor|Primary School|
> |Age Group|20-30|20-30|10-20|80-90|50-60|50-60|20-30|10-20|
> |Expertise|Finance|Psychology|N/A|Business|Literature|Finance|Media|N/A|
>
> ||P1|P2|P3|P4|P5|P6|P7|P8|Overall|
> |------------|:------:|:------:|:------:|:------:|:------:|:------:|:------:|:------:|:--------:|
> |Pass@1 (%)|81.5|92.6|88.9|66.7|92.6|85.2|85.2|88.9|85.2|
>
> For the families highlighted by the reviewer, the current study suggests these families are human-solvable, although several require longer completion times and therefore raise usability trade-offs. In particular, the average completion time is 28.2s for Illusory Ribbons and 26.5s for Structure From Motion. Within the Spooky family, completion times are generally moderate, with Spooky Jigsaw (71.8s) being the main exception. The other Spooky variants all have average completion times under 20s. More broadly, the overall average completion time across all 268 human trials is 31.1s per puzzle. Although a few families are relatively more time-consuming, such as 3D Viewpoint (120.8s), Spooky Jigsaw (71.8s), Subway Paths (71.4s), and Multi Script (63.1s), the large majority of families are completed within one minute on average. Overall, these results suggest that human participants can solve most task families reliably after deliberation, with the main difference being time rather than solvability. We will add a related discussion in the revised version.
>
> >**Q1: Long-term robustness under scalable CAPTCHA generation.**
>
> Thanks for the suggestion. We clarify that scalable synthesis alone will not yield permanent security. In our view, procedural generation plays a narrower but still useful role: it enables continual refresh, reduces memorization/replay value, and makes static data collection attacks less useful. Long-term robustness must still come from targeting failure modes that remain disproportionately costly for agents under realistic browser interaction, especially temporal evidence integration, latent-state tracking, and perception-to-action alignment, together with deployment controls such as server-side verification, rate limiting, and anomaly-based detection. So, the right objective is not to "freeze" agent capability forever, but to maintain a moving-target defense that preserves human usability while forcing attackers into increasingly brittle and expensive attack workflows. We will make this discussion and positioning more explicit in the revision's Impact Statement Section.
>
> >**Q2: Whether user-authorized agents should be treated as bots.**
>
> Good question. We view this primarily as a policy and deployment question rather than a broad general claim. A user-authorized agent is not necessarily malicious, and assistive or accessibility-oriented agent use may be entirely legitimate. Our threat model is narrower: workflows where the site explicitly wishes to verify direct human presence or prevent scalable automated execution (e.g., abuse-prone account creation, scraping-sensitive endpoints, or fraud-sensitive actions).
>
> In those settings, an agent, even if delegated by a user, can still fall outside the intended access model, so a CAPTCHA-style defense may remain appropriate. At the same time, assistive/authorized agents should not be treated the same as malicious automation, and we will clarify this distinction in the revision.
>
> >**Q3: Typo at Line 266.**
>
> Thanks for pointing this out. We will carefully check the whole paper again and correct all typos in the revised version.

---

> > ### Author Rebuttal · Reviewer_Uzom · 2026-04-01
> >
> > Thanks for providing the additional human study results. As a result, I will raise my score.

---

> > > ### Author Response · Authors · 2026-04-02
> > >
> > > Thanks and we are very glad to hear this. We will incorporate all these additional experiments, results, and clarifications into the revised paper. Please feel free to let us know if you have any further questions.

---

### Decision · Program_Chairs · 2026-04-30

**Decision:**

Accept (regular)

**Comment:**

**Summary:**

The paper argues that traditional CAPTCHAs are no longer effective against modern GUI agents and proposes a new set of interactive CAPTCHAs based on gaps between human and AI behavior. Rather than making puzzles harder in the usual way, the design targets areas where agents still struggle, such as perception, memory, and low-level actions, while keeping the tasks easy for humans. The results suggest these CAPTCHAs are much more difficult for current agents and still practical for human users.

Strengths and weaknesses from initial reviews:

**Strengths:**

1. This is a timely and important problem, and the paper is well motivated.
2. The proposed CAPTCHA families are diverse, thoughtfully designed, and supported by a useful taxonomy of human–AI gaps.
3. The experiments are strong overall, with clear results, ablations, and useful analysis of cost and attack difficulty.
4. The work is practical: rule-based verification avoids the need for manual labeling, and the overall framework could be useful beyond the specific tasks tested.

**Weaknesses:**
1. The biggest concern is the human study: the paper claims high human success, but the evaluation details are not clear enough.
Some tasks may also be difficult for real users, especially people without technical backgrounds or with weaker visual/spatial skills.
2. A few tasks may require better internet or smoother interaction than standard CAPTCHAs.
3. The paper does not test adaptive attackers, so it is still unclear how well the approach will hold up as AI systems improve.
4. The formal modeling adds limited value, since it does not lead to strong theory or guarantees.

All reviewers are happy with the rebuttal and recommend accept.